# Clinical outcomes of adults hospitalized for laboratory confirmed respiratory syncytial virus or influenza virus infection

Magdalena Chorazka[1], Domenica Flury[2], Kathrin Herzog[3], Werner C. Albrich[2], Danielle Vuichard-Gysin[1,4]*

1 Department of Internal Medicine, Cantonal Hospital Muensterlingen, Thurgau Hospital Group, Muensterlingen, Switzerland, 2 Department of Infectious Diseases and Hospital Epidemiology, Cantonal Hospital St. Gallen, St. Gallen, Switzerland, 3 Department of Clinical Microbiology, Thurgau Hospital Group, Muensterlingen, Switzerland, 4 Department of Infectious Diseases and Hospital Epidemiology, Cantonal Hospital Muensterlingen, Thurgau Hospital Group, Muensterlingen, Switzerland

* danielle.vuichard-gysin@stgag.ch

## Abstract

### Objectives

Respiratory syncytial virus (RSV) can cause severe disease in adults, but far less is known than for influenza. The aim of our study was to compare the disease course of RSV infections with influenza infections among hospitalized adults.

### Methods

We retrieved clinical data from an ongoing surveillance of adults hospitalized with RSV or influenza virus infection in two acute care hospitals in North-Eastern Switzerland during the winter seasons 2017/2018 and 2018/2019. Our main analysis compared the odds between RSV and influenza patients for admission to an intensive care unit (ICU) or in-hospital death within 7 days after admission.

### Results

There were 548 patients, of whom 79 (14.4%) had an RSV and 469 (85.6%) an influenza virus infection. Both groups were similar with respect to age, sex, smoking status, nutritional state, and comorbidities. More RSV patients had an infiltrate on chest radiograph on admission (46.4% vs 29.9%, p = .007). The proportion of patients with RSV who died or were admitted to ICU within seven days after admission was 19.0% compared to 10.2% in influenza patients (p = .024). In multivariable analysis, a higher leukocyte count (adjusted OR 1.07, 95% CI 1.02–1.13, p = .013) and the presence of a pneumonic infiltrate (aOR 3.41, 95% CI 1.93–6.02) significantly increased the risk for experiencing the adverse primary outcome while the effect of the underlying viral pathogen became attenuated (aOR 1.18, 95% CI 0.58–2.41, p = .0655).

**Data Availability Statement:** The data underlying this study have been uploaded as Supporting Information and to https://osf.io/7udb5/.

**Funding:** The authors received no specific funding for this work.

**Competing interests:** The authors have declared that no competing interests exist.

## Conclusions

Our results suggest that RSV is responsible for clinical courses at least as severe as influenza in adults. This supports the need for better guidance on diagnostic strategies as well as on preventive and therapeutic measures for hospitalized adults with RSV infection.

## Introduction

Viral respiratory infections are one of the most common reasons for outpatient consultation and hospitalization every year between October and April [1, 2]. They are associated with increased morbidity and mortality either caused by the virus itself, due to bacterial superinfection, or deterioration of already existing chronic medical conditions [3, 4]. Symptoms and signs of influenza overlap with those seen with other respiratory pathogens, including RSV [5, 6]. The burden of influenza has been characterized best compared to other viral respiratory infections and has a long tradition of epidemiological surveillance [7]. In addition, diagnostic procedures, preventive measures such as vaccination and isolation precautions, and treatment with neuraminidase inhibitors have been well investigated [8]. For years, viral respiratory infections were dichotomized in clinical practice into influenza and other influenza-like respiratory tract infections which ignored the potential relevance of RSV, mostly because of the absence of reliable diagnostic tests and therapeutic options [9]. Only recently, RSV has been recognized as important risk factor for hospitalization and mortality in the elderly population [4, 10–12]. In addition, patients with underlying cardiopulmonary diseases seem to have a greater risk for symptomatic RSV infection and subsequent healthcare utilization [13]. There are no approved vaccines or direct treatment options for non-immunocompromised adults yet. Therefore, routine diagnostics and isolation precautions are the only measure that can be implemented to potentially reduce nosocomial spread of RSV [14]. In Switzerland, there is a well-established influenza surveillance for hospitalized patients spanning over several years and a systematic COVID-19 surveillance in patients admitted to hospital has been rapidly established [15]. In contrast, data regarding the epidemiology and outcomes of hospitalized adults with RSV infection in Switzerland are scarce. Therefore, the proportion of RSV infections in adult patients admitted to acute care, their clinical course and outcome remain ill-defined. In addition, recommendations on routine diagnostics or on isolation precautions for non-immunocompromised adult patients are lacking. If clinical course and outcomes of RSV and influenza virus infections are comparable, consideration should be given to establish a similar prospective surveillance in adults admitted to hospital to better inform clinicians and public health authorities. Our study goal therefore was to compare the clinical burden of RSV and influenza virus infections among adult patients hospitalized for acute respiratory illness in two acute care hospitals in Northeastern Switzerland.

## Methods

### Study design

We performed a retrospective analysis from patients with laboratory confirmed influenza or RSV infection, who were admitted to one of the two acute care hospitals in the canton Thurgau during winter season I (2017/2018) or II (2018/2019).

### Study setting and population

The Thurgau hospital group has about 570 acute care beds. To prevent the spread of influenza in the hospital, the division of infection control established a new surveillance program in

October 2017 for patients who are being hospitalized with influenza or RSV infection. At this time, the microbiology laboratory introduced a combined RT-PCR for the diagnosis of influenza and RSV infection. Clinicians were advised to obtain a nasopharyngeal swab for microbial diagnosis of influenza or RSV as a standard diagnostic procedure in all patients with symptoms and signs of acute respiratory infection (ARI) and whose condition required hospital admission. Patients with ARI presented with an influenza-like illness consisting of fever, general malaise, cough, or myalgia or had a chest infiltrate suggestive for viral pneumonia. Cases on the wards who developed new ARI also underwent testing for influenza and RSV. All patients who tested positive for influenza were immediately placed in droplet isolation as per local infection control guideline.

## Data collection

The hospital influenza and RSV surveillance served as basis for our study. As part of this surveillance program, the division of infection control routinely collects data on microbiological diagnosis, length of hospital stay, admission to ICU, and death during hospital stay in patients who test positive for influenza virus or RSV by means of an RT-PCR. This prospective surveillance starts in calendar week 44 in the previous year and extends to calendar week 16 in the following year.

We included all adult patients (equal or older than 18 years of age at time of hospitalization) who were hospitalized during influenza seasons I and II with a laboratory confirmed influenza or RSV infection. Only the first episode was considered. We excluded patients with an RSV/influenza co-infection. Additional data on patient demographics, comorbidities, vital signs on admission, laboratory results, treatment and outcomes were retrospectively retrieved from individual electronic patient files. These data were entered into the electronic patient file during routine patient care. Every death in the community is automatically reported to the public hospitals in the canton and is therefore visible in the electronic health record. This enabled us to also determine the 30-day mortality.

## Microbiological diagnostic

Our laboratory has introduced the Xpert Flu/RSV Xpress Assay, which is a rapid, automated in vitro diagnostic test for qualitative detection and differentiation of influenza A and B viruses and RSV performed on the Cepheid GeneXpert Xpress System. The Xpert Flu/RSV Xpress Assay has consistently shown excellent performance [16, 17]. In a recent prospective evaluation, the sensitivity was almost 100% for influenza A and B and 90.5% for RSV, and the specificity ranged from 98.6% to 99.7% for influenza A, influenza B, and RSV, respectively. The positive predictive value of the Xpert assay is 95.5% for Influenza A, 97.0% for Influenza B, and 90.5% for RSV [18]. Results are automatically sent to the local infection prevention and control team for continuous surveillance.

## Statistical analysis

Our primary endpoint was a composite of ICU-admission (need for ventilatory or vasopressor support) and/or in-hospital death within 7 days after hospital admission. We deemed 7 days an appropriate duration of follow-up since we expected a high discharge rate for those with a favourable outcome after this period. In addition, a previous meta-analysis on the burden of RSV in older adults found that hospital stays were rather short (< 7 days) and a high proportion of ICU admission occurred within this time frame [19]. A longer observation period could lead to overestimation of the frequency of the primary endpoint, with the additional disadvantage that a direct association of the outcome with the initial RSV or influenza virus

infection will become less likely. Secondary outcomes included cardiopulmonary and cardio-vascular events, proportion and duration of antibiotic treatment, discharge to nursing home, 30-day re-admission and 30-day all-cause mortality.

For descriptive analyses we used medians and interquartile range (IQR) and numbers along with percentages as appropriate. We applied Chi-square test and the Mann-Whitney-U test for comparison of proportions and medians, respectively. We used binary logistic regression to model the crude prediction of RSV and influenza virus infection for the probability of ICU admission or death within the first 7 days after hospital admission. We then applied multivariable models to adjust for additional predictors of our primary outcome in patients with RSV compared to influenza virus infection. Only predictors that were statistically significant in univariate analyses were considered in the multivariable model. A p-value of < 0.05 was considered statistically significant. All analyses were performed using SPSS Version 27.0 [20].

### Ethical approval

The ethics commission of North-Eastern Switzerland approved the study (No. 2019–01423). The ethics committee granted our waiver request to obtain informed consent from individual patients because of the retrospective nature of the work and the large sample size.

## Results

Over the two winter seasons there were a total of 1983 patients hospitalized for ARI with a nasopharyngeal swab sample available that had been analyzed for influenza virus and RSV. Of these, 79 (4.0%) tested positive for RSV and 469 (23.7%) tested positive for influenza. There were significant differences in the proportion of admissions for RSV and influenza comparing the two respiratory seasons (Table 1, p < .001). The cumulative numbers of laboratory confirmed RSV and influenza cases per calendar week are shown in Fig 1.

The two groups did not differ significantly in age, gender, body mass index (BMI), smoking habits, nutritional status, or comorbidities. Median age was 78 years (IQR 65–84) and 74 years (63–83) in RSV and influenza patients, respectively. In both groups, there were slightly more women (60.8% of RSV patients and 57.4% of influenza patients, p = .571). The most frequent comorbidities were hypertension and congestive heart failure. We found a statistically significant difference in clinical signs and symptoms between the two groups with a higher median temperature of 38.0˚C versus 37.4˚C (p = .019) and a lower median leukocyte count of 7.2 x 10^9/L versus 9.2 x 10^9/L (p < .001) in the influenza group compared to the RSV group. In contrast, the proportion of patients with an infiltrate on chest X-ray was significantly higher in the RSV group (46.4% vs. 29.9%, p = .007) (Table 1).

With 19% (15/79) in the RSV group versus 10.2% (48/469) in the influenza group, the proportion of patients with RSV reaching our primary endpoint compared to patients with influenza was significantly higher (p = .024). The proportion of patients reaching any of our prespecified secondary end points, such as need for ventilatory support, development of congestive heart failure, length of hospital stay, re-admission within 30 days after discharge or death of any cause after 30 days were equal or higher in the RSV group compared to influenza. However, none of these differences in proportions were statistically significant (Table 2).

In the crude analysis, RSV patients were twice as likely to be admitted to ICU or die within 7 days after admission compared to patients with influenza (OR 2.06, 95% CI 1.09–3.90, p = .027) (Table 3). However, the association was no longer significant after controlling for temperature, leukocyte count, and presence of a pulmonary infiltrate on admission. While neither fever nor the underlying respiratory virus appeared to have a significant impact on the primary outcome in the adjusted analysis, a higher leukocyte count (adjusted OR 1.07, 95% CI 1.02–

**Table 1. Characteristics of patients with laboratory confirmed RSV or influenza virus infection on admission.**

| | RSV | | Influenza | | p-value [a] |
|---|---|---|---|---|---|
| Total cases, n (%) | 79 | (14.4) | 469 | (85.6) | |
| Admitted during season I | 27 | (34.2) | 270 | (57.6) | <**0.001** |
| Admitted during season II | 52 | (65.8) | 199 | (42.4) | |
| Age, median (IQR) | 78 | (65–84) | 74 | (63–83) | 0.072 |
| Age 18–65 years, n (%) | 21 | (26.6) | 146 | (31.1) | |
| Age 65–80 years, n (%) | 25 | (31.6) | 160 | (34.1) | 0.470 |
| Age > 80 years, n (%) | 33 | (41.8) | 163 | (34.8) | |
| Females, n (%) | 48 | (60.8) | 269 | (57.4) | 0.571 |
| Ever smoked, n (%) | 32 | (40.5) | 195 | (41.6) | 0.680 |
| BMI, median (IQR) | 27.0 | (22.0–32.0) | 26.0 | (22.4–29.1) | 0.280 |
| Admitted from home, n (%) | 67 | (84.8) | 408 | (87.0) | |
| Admitted from nursing home, n (%) | 9 | (11.4) | 49 | (10.4) | 0.800 |
| Admitted from other healthcare facility, n (%) | 3 | (3.8) | 12 | (2.6) | |
| Nutritional risk score, median (IQR) | 2 | (1–2) | 2 | (1–3) | 0.690 |
| **Comorbidities** | | | | | |
| Myocardial infarction [b] | 3 | (3.8) | 8 | (1.7) | 0.380 |
| Congestive heart failure [b] | 38 | (48.1) | 199 | (42.4) | 0.347 |
| Hypertension | 47 | (59.5) | 266 | (56.7) | 0.644 |
| Stroke [b] | 2 | (2.5) | 11 | (2.3) | 1.000 |
| COPD | 17 | (21.5) | 87 | (18.6) | 0.534 |
| Asthma | 7 | (8.9) | 29 | (6.2) | 0.374 |
| Diabetes | 10 | (12.7) | 73 | (15.6) | 0.505 |
| Hematological disorder | 5 | (6.3) | 12 | (2.6) | 0.083 |
| Charlson Comorbidity Index, median (IQR) | 3 | (1–5) | 3 | (1–5) | 0.363 |
| **Clinical features on admission** | | | | | **p-value [a]** |
| Temperature (in ˚C), median (IQR) | 37.4 | (37.0–38.0) | 38.0 | (37.0–38.6) | **0.019** |
| Blood pressure systolic, median (IQR) | 132 | (120–152) | 134 | (119–148) | 0.958 |
| Peripheral O2 saturation (in %), median (IQR) | 95 | (91–97) | 95 | (92–98) | 0.152 |
| CRP (in mg/L), median (IQR) | 31.0 | (11–84.5) | 38.0 | (16–93.3) | 0.277 |
| Leukocyte count (in 10e9/L), median (IQR) | 9.2 | (7.0–11.7) | 7.2 | (5.2–10.0) | <**0.001** |
| Infiltrate on chest radiograph | 32 | (40.5) | 115 | (24.5) | **0.007** |

[a] p-values were calculated using Chi-square for comparison of proportions and Mann-Whitney U-test for comparison of medians
[b] within 1 year prior to admission

1.13) and the presence of a pulmonary infiltrate on chest X-ray (aOR 3.41, 95% CI 1.93–6.02, p < .001) appeared to significantly increase the risk of an unfavourable outcome. Patients with pneumonia on admission were 3.41 times more likely to be transferred to ICU or to die within the first 7 days than patients without pneumonia. The addition of this predictor to the model, however, seems to attenuate the effect of RSV as indicated by a decrease in the adjusted OR.

To better understand the association of a pulmonary infiltrate with clinical outcome, we performed a subgroup analysis, examining the impact of age on ICU admission or death within 7 days after admission in patients presenting with a pulmonary infiltrate stratified by influenza and RSV. First, we found that 29 of 115 (25.2%) influenza cases and 7 of 32 (21.9%) RSV cases in this subgroup experienced the primary outcome. The difference was not statistically significant. However, we note that the proportion of cases reaching the outcome between the influenza and RSV group clearly differed by age category with a higher proportion of

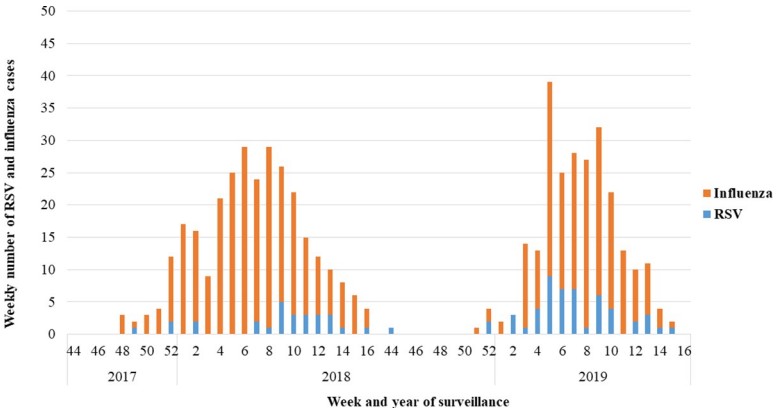

**Fig 1. Weekly distribution of confirmed RSV and influenza virus infections 2017–2019.** The surveillance started in week 44 in the previous year and ended in week 16 in the following year).

elderly (over 80 years of age) patients in the RSV group (Fig 2). The identified virus modified the effect of age on the development of the outcome of ICU admission or death among those with a pulmonary infiltrate.

## Discussion

This is the first cohort study in Switzerland that compares clinical characteristics and outcomes of adult patients being hospitalized with PCR-confirmed RSV or influenza virus infection. Our results are compatible with previous findings from other countries, which suggest that RSV infections in hospitalized patients can cause a similar or even more severe disease

**Table 2. Outcomes of patients with laboratory confirmed RSV and influenza virus infection.**

|  | RSV n = 79 | | Influenza n = 469 | | p-value [a] |
|---|---|---|---|---|---|
| **Primary outcome** | | | | | |
| ICU admission or death within 7 days, n (%) | 15 | (19.0) | 48 | (10.2) | **0.024** |
| **Secondary outcomes** | | | | | |
| ICU admission, n (%) | 13 | (16.5) | 44 | (9.4) | 0.057 |
| ICU admission within 7 days, n (%) | 12 | (15.2) | 37 | (7.9) | **0.035** |
| Ventilatory support, n (%) | 6 | (7.6) | 24 | (5.1) | 0.419 |
| Hospital-acquired pneumonia, n (%) | 10 | (12.7) | 54 | (11.5) | 0.770 |
| Congestive heart failure, n (%) | 16 | (20.3) | 73 | (15.6) | 0.296 |
| Death within 7 days (%) | 4 | (5.1) | 8 | (1.7) | 0.059 |
| Death 30 days after admission | 8 | (10.1) | 34 | (7.2) | 0.374 |
| Total length of stay, median (IQR) | 6 | (4–12) | 6 | (3–10) | 0.348 |
| Discharge into nursing homes, n (%) | 3 | (4.7) | 27 | (6.7) | 0.605 |
| Re-admission within 30 days after discharge, n (%) | 9 | (12.5) | 43 | (9.7) | 0.466 |
| Total duration of antibiotic therapy in days, median (IQR) | 2 | (0–6) | 0 | (0–6) | 0.287 |
| Any antibiotic treatment, n (%) | 42 | (53.2) | 212 | (45.2) | 0.223 |
| **ICU or death within 7 days according to season** | **RSV** | | **Influenza** | | p-value [a] |
| Admitted in season I, n of total (%) | 5/27 | (18.5) | 25/270 | (9.3) | 0.128 |
| Admitted in season II, n of total (%) | 10/52 | (19.2) | 23/199 | (11.6) | 0.145 |

[a] p-values were calculated using Chi-square for comparison of proportions.

**Table 3. Univariate and multivariable analysis for the probability of ICU admission or death within 7 days after hospital admission in patients with RSV compared to influenza virus infection.**

|  |  | OR (95% confidence interval) | | p-value |
|---|---|---|---|---|
| **Unadjusted** | RSV infection | 2.06 | (1.09–3.90) | **0.027** |
| **Adjusted** | RSV infection | 1.18 | (0.58–2.41) | 0.655 |
|  | Temperature (in˚C) | 0.91 | (0.69–1.20) | 0.489 |
|  | Leukocyte count (in 10e9/L) | 1.07 | (1.02–1.13) | **0.013** |
|  | Pulmonary infiltrate | 3.41 | (1.93–6.02) | <0.001 |

than seasonal influenza in non-immunocompromised adult patients [4, 10, 21]. Overall, it seems, that a higher leukocyte count and the presence of an infiltrate increase the risk of an adverse outcome. There was a statistically significantly higher proportion of leukocytosis among patients with RSV compared to those with influenza, albeit with most patients having only mild leukocytosis at most. In multivariable analysis, leukocytosis but not RSV was associated with more severe clinical course. The association between RSV and leukocytosis has received some attention recently. A study of 243 US adults with RSV showed that 11% had leukocytosis with no difference between those with moderate to severe disease (10%) and those with milder illness (12%, p = 0.8). In this study, patients with more severe disease had more frequently pneumonia [22]. In a large Chinese study of 1046 adults with RSV, 38% had leukocytosis overall including 44% of those with influenza coinfection, while even with these large case numbers the authors were unable to find an association between leukocytosis and severity of radiologic infiltrates [23]. In children with RSV, one study found a leukocytosis in 24% of those with fever and 18% of those without fever. In those with fever, there was also a higher likelihood of bacterial infection, but among febrile children with a white blood count (WBC) below 30,000 per milliliter, there was no difference in the WBC between those with and those without bacterial superinfection [24]. Therefore, it remains speculative, whether the observed higher WBC count in patients with RSV compared to influenza in our study is related to more frequent bacterial superinfections (antibiotic therapy had a non-significantly higher point estimate in RSV than influenza) or is a marker of more severe clinical presentation of a viral infection. WBC was previously shown to have similar prognostic accuracy as CRP for severe outcome in community acquired pneumonia but without being sensitive or specific enough to work as a predictor on its own [25].

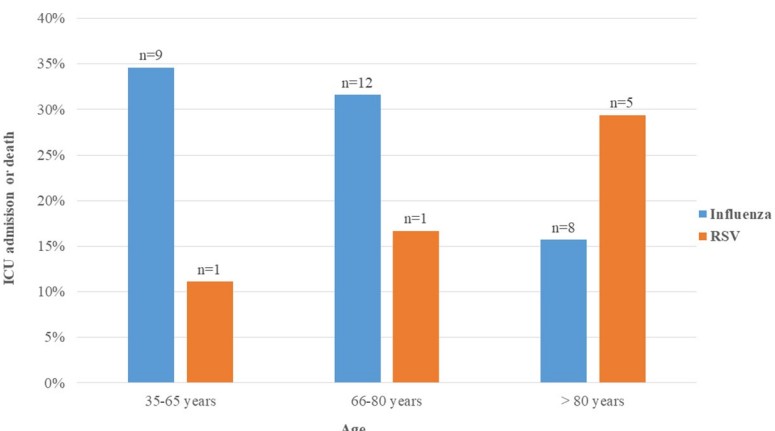

**Fig 2. Proportion of patients with RSV or influenza virus infection and a pulmonary infiltrate on admission reaching the primary endpoint according to age category.**

In contrast to previous results that pointed towards older patients and patients with comorbidities [4, 10], our primary findings of serious outcomes in patients with RSV were independent of age and comorbidities. We therefore further explored the potential influence of age in those patients with a pulmonary infiltrate on admission. When examining this subgroup for a possible influence of age, we found a higher proportion of RSV infected individuals over 80 years of age who reached the primary endpoint compared to younger age groups. In contrast, there was an opposite trend among influenza-infected individuals rendering the type of respiratory virus an effect modifier for the presence of a pulmonary infiltrate. However, the numbers were too small to perform any meaningful statistics. This result therefore needs to be examined in more detail in a larger cohort, ideally using virus typing and looking at the two endpoints separately.

In addition, none of our secondary prespecified endpoints showed a statistically significant difference between the two groups, suggesting that the severity of the disease at the time of admission, rather than the virus itself was responsible for a poorer outcome. These findings are important because in Switzerland, during routine care, RSV is still rarely considered an important pathogen in non-immunocompromised adults. Our data therefore underline the importance of developing uniform guidelines on diagnosis, prevention, and control of RSV in adults and lay the basis of treatment criteria when effective therapies become available. Given the considerable overlap between underlying comorbidities and clinical manifestations and different management, virological diagnosis will likely become even more important in the future.

One strength of our study was the systematic case finding by means of combined influenza and RSV molecular testing, thereby minimizing the risk of selection bias. Furthermore, thanks to our comprehensive analysis of clinical characteristics and various important endpoints, we were able to show that RSV is comparable to influenza in several respects.

However, our study has also some limitations. First, as it is the case in most retrospective cohort studies, there is a certain risk of referral bias. Since we were unable to evaluate for the duration of symptoms prior to admission, we cannot be sure, whether patients presenting with more classical symptoms suggesting influenza were referred to the hospital at an earlier stage and, therefore, rendered them more likely to have a favorable outcome. Second, because of the limited number of hospitalized RSV patients, the number of events related to our primary end point was small, which limited the number of confounding factors that could be adjusted for. Nonetheless, our results highlight that although the population burden of hospitalization is higher for influenza than for RSV, the individual risk of a severe outcome may be higher for RSV than for influenza.

Our study was designed before the SARS-CoV-2 pandemic. This changed situation emphasizes even more the need to not neglect other respiratory viruses such as RSV, whose course can be at least as severe as influenza and which cannot readily be distinguished based on clinical grounds alone [26]. The transmissibility of a pathogen and severity of the causing disease must be considered when implementing additional transmission-based precautions. RSV transmission risk has rarely been studied in non-severely immunocompromised adults in healthcare settings, but from what is known, this risk may be substantial [14]. In the absence of effective treatment and vaccines, nursing homes and acute care settings should be conscious to take additional preventive measures for all respiratory infections.

In Switzerland, a prospective multicenter study of hospitalized influenza patients to evaluate the effectiveness of infection prevention measures has been initiated in 2015 [27]. Since this year, a national epidemiological surveillance on SARS-CoV-2 in hospitalized patients is also in place; and only recently, influenza virus infection was added to this surveillance [28]. With the growing recognition that RSV can cause a similarly severe course as influenza,

emphasis on prevention in and out of the hospital, future intervention studies and prospective surveillance systems should include RSV patients.

## Conclusion

Our data provide further evidence that RSV infection is a serious problem in hospitalized adults, even in those without severe immunosuppression. RSV infection should therefore be diagnosed and further transmission, especially in healthcare facilities, must be prevented by appropriate interventions.

## Supporting information

**S1 Data.**
(XLSX)

## Author Contributions

**Conceptualization:** Magdalena Chorazka, Domenica Flury, Werner C. Albrich, Danielle Vuichard-Gysin.

**Data curation:** Danielle Vuichard-Gysin.

**Formal analysis:** Danielle Vuichard-Gysin.

**Methodology:** Magdalena Chorazka, Kathrin Herzog, Werner C. Albrich, Danielle Vuichard-Gysin.

**Supervision:** Werner C. Albrich, Danielle Vuichard-Gysin.

**Writing – original draft:** Magdalena Chorazka.

**Writing – review & editing:** Domenica Flury, Kathrin Herzog, Werner C. Albrich, Danielle Vuichard-Gysin.

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
