## [Decision Letter · Decision Letter 0]

6 Mar 2021

PONE-D-21-03134

Clinical outcomes of adults hospitalized for laboratory confirmed respiratory syncytial virus or influenza virus infection .

PLOS ONE

Dear Dr. Vuichard-Gysin,

Thank you for submitting your manuscript to PLOS ONE. After careful consideration, we feel that it has merit but does not fully meet PLOS ONE’s publication criteria as it currently stands. Therefore, we invite you to submit a revised version of the manuscript that addresses the points raised during the review process.

We look forward to receiving your revised manuscript.

Kind regards,

Tai-Heng Chen, M.D.

Academic Editor

PLOS ONE

Reviewers' comments:

Reviewer's Responses to Questions

**Comments to the Author**

1. Is the manuscript technically sound, and do the data support the conclusions?

Reviewer #1: Partly

Reviewer #2: Yes

Reviewer #3: Partly

2. Has the statistical analysis been performed appropriately and rigorously? 

Reviewer #1: No

Reviewer #2: Yes

Reviewer #3: Yes

3. Have the authors made all data underlying the findings in their manuscript fully available?

Reviewer #1: Yes

Reviewer #2: Yes

Reviewer #3: Yes

4. Is the manuscript presented in an intelligible fashion and written in standard English?

Reviewer #1: Yes

Reviewer #2: Yes

Reviewer #3: Yes

5. Review Comments to the Author

Reviewer #1: This is an interesting piece of research that has not been explored well enough. However, I have a few questions and it relates mainly toward the small number of events and the analysis.

1. The outcomes table should be placed in the main manuscript instead of the supplementary material.

2. I’m trying to understand how the data has been collected as the study design section mentioned “retrospective analysis” and study population and data collection section writes “prospectively collected data”. Were these patients prospectively recruited or were cases identified through the hospital surveillance program? If the latter, would it have been possible to increase the sample size to another season as the number of events of the RSV groups is very low (n=15).

3. Can the authors elaborate on how the cases with ARI were identified? Were they based on ED diagnosis codes? In addition, I see that all 588 patients had a chest radiograph. Is that a requirement for all ARI admissions?

4. The follow-up time here is only 7 days and likely to encounter ties in their analysis. Can the authors explain the rationale in performing a PH model instead of a simpler logistic regression model?

5. The authors adjusted for age and comorbidities. In my opinion, that might not be necessary since they were found to be insignificant in Table 1. Why was adjustment only done for pulmonary infiltrate but not the other clinical features found significant on admission in Table 1?

6. Presence of pulmonary infiltrate has been found to increase the risk of ICU admission or mortality. This was presented as a significant finding and discussed briefly. However, this finding does not seem to align with the objectives of the study of comparing disease severity of RSV vs influenza.

In the discussion, the authors “suspect the influence of pneumonic infiltrate was so strong that age and comorbidities were no longer relevant..”. From Table 2, it looks like the effects of pneumonic infiltrate attenuating the effect of the RSV instead of these potential confounders. The authors could relate these findings better by further investigating the proportion of patients who experienced the event and had pulmonary infiltrates , stratified by RSV/influenza.

Reviewer #2: Minor Comments:

Methods:

• The influence of seasonality (season I vs. season II) on the outcome of the patients is not mentioned. It would be interesting to explore if the RSV/Influenza-associated hospitalizations varied substantially during the study period, related to the particular virus types and subtypes in circulation

Results:

• The authors stated that the proportion of patients with infiltrate on chest X-ray was significantly higher in the RSV group (46.4% vs. 29.9%). However, figure 1 shows 40.5% vs.24.5%. Please revise

• Primary endpoint is part of the main objective of this paper but the results are not shown in main body. Please move Suppl. Table 1. to main body

• Justify why the covariates tested in the multivariate model were selected (e.g. Significance criteria/supported by previous literature)

Reviewer #3: 1. As the inpatient follow up was limited to 7 days, would the authors be expected to fully describe the clinical outcome of adults patients hospitalised with RSV or influenza infection? This is important as "outcomes" is present in both the full and short title of the manuscript.

2. Is Figure 2 necessary? In its current form, it appears to add little to the manuscript as the cutoff is at 7 days

3. Please review reference 16 - this research article had described workflow parameters and in its results, it did not describe sensitivity or specificity of the test referred to.

4. Would the authors consider expanding their discussion to other points including clinical prediction models for RSV, possible preventive measures, nosocomial transmission, possible planned clinical interventions for RSV.

6. PLOS authors have the option to publish the peer review history of their article (what does this mean?). If published, this will include your full peer review and any attached files.

Reviewer #1: No

Reviewer #2: No

Reviewer #3: No

---

## [Author Response · Author response to Decision Letter 0]

16 Apr 2021

We would like to thank all reviewers for their time and scrutiny to improve this manuscript. We hope that our responses will be satisfying. Please find below our point-by-point reply:

Reviewer #1:

This is an interesting piece of research that has not been explored well enough. However, I have a few questions and it relates mainly toward the small number of events and the analysis.

1. The outcomes table should be placed in the main manuscript instead of the supplementary material.

Response: This is an excellent suggestion, and we would like to thank you for that. We moved the outcomes table in the main manuscript.

2. I’m trying to understand how the data has been collected as the study design section mentioned “retrospective analysis” and study population and data collection section writes “prospectively collected data”. Were these patients prospectively recruited or were cases identified through the hospital surveillance program? If the latter, would it have been possible to increase the sample size to another season as the number of events of the RSV groups is very low (n=15).

Response: Thank you for raising this question which gives us the opportunity to provide some clarifications. The study population was indeed identified through the hospital surveillance program. Since the hospital’s microbiology laboratory introduced combined PCR testing for RSV and influenza only at the beginning of October 2017, these are all available data. Based on your comment that it is not entirely clear how the data were collected, we have taken the liberty of revising the methodology section and hope that it is more understandable now.

The section now reads as follows (page 4, lines 82-102 and page 5, lines 103-113):

Study design

We performed a retrospective analysis from patients with laboratory confirmed influenza or RSV infection, who were admitted to one of the two acute care hospitals in the canton Thurgau during winter season I (2017/2018) or II (2018/2019). 

Study setting and population

The Thurgau hospital group has about 570 acute care beds. To prevent the spread of influenza in the hospital, the division of infection control established a new surveillance program in October 2017 for patients who are being hospitalized with influenza or RSV infection. At this time, the microbiology laboratory introduced a combined RT-PCR for the diagnosis of influenza and RSV infection. Clinicians were advised to obtain a nasopharyngeal swab for microbial diagnosis of influenza or RSV as a standard diagnostic procedure in all patients with symptoms and signs of acute respiratory infection (ARI) and whose’ condition required hospital admission. Patients with ARI presented with an influenza-like illness consisting of fever, general malaise, cough, or myalgia or had a chest infiltrate suggestive for viral pneumonia. Patients hospitalized on the wards who developed new ARI also underwent testing for influenza and RSV. All patients who tested positive for influenza were immediately placed in droplet isolation as per local infection control guideline. 

Data collection

The hospital influenza and RSV surveillance served as basis for our study. As part of this surveillance program, the division of infection control routinely collects data on microbiological diagnosis, length of hospital stay, admission to ICU, and death during hospital stay in patients who test positive for influenza virus or RSV by means of an RT-PCR. This prospective surveillance starts in week 44 in the previous year and extends to week 16 in the following year. 

We included all adult patients (equal or older than 18 years of age at time of hospitalization) who were hospitalized during influenza seasons I and II with a laboratory confirmed influenza or RSV infection. Only the first episode was considered. We excluded patients with an RSV/influenza co-infection. Additional data on patient demographics, comorbidities, vital signs on admission, laboratory results, treatment and outcomes were retrospectively retrieved from individual electronic patient files. These data were entered into the electronic patient file during routine patient care. Every death in the community is automatically reported to the public hospitals in the canton and is therefore visible in the electronic health record. This enabled us to also determine the 30-day mortality.

3. Can the authors elaborate on how the cases with ARI were identified? Were they based on ED diagnosis codes? In addition, I see that all 588 patients had a chest radiograph. Is that a requirement for all ARI admissions?

Response: We are grateful for this question. Before introducing the combined PCR, clinicians were instructed about the clinical and radiological criteria they should apply to decide whether a patient qualifies for a nasopharyngeal swab. However, whether a patient was tested by PCR eventually remained at the discretion of the responsible physicians working in the ED. The majority of patients with new clinical signs of ARI receive at least a chest X-ray. In this cohort, 454 of 548 (82.8%) received a chest x-ray. Since the question relates to the previous one, we kindly ask you to review the revisions already mentioned above under point 2. We hope that our revision will sufficiently answer your question.

4. The follow-up time here is only 7 days and likely to encounter ties in their analysis. Can the authors explain the rationale in performing a PH model instead of a simpler logistic regression model?

Response: We would like to thank the reviewer for this important question that enables us to improve the statistical analyzes and draw correct inferences. We agree that we would need to consider ties in our Cox PH regression model. This could be done by an Efron or Breslow approximation. However, instead of performing a more complex analysis that would be less understandable we decided to redo our main analysis by applying simple binary logistic regression. With the new analysis, main outcomes and inferences that can be drawn remain the same. We kindly ask you to refer to our response under the point 5 to review all changes that have been made to the manuscript.

5. The authors adjusted for age and comorbidities. In my opinion, that might not be necessary since they were found to be insignificant in Table 1. Why was adjustment only done for pulmonary infiltrate but not the other clinical features found significant on admission in Table 1?

Response: Again, we are grateful for raising this important issue. Being in line with your previous suggestion to rather stick with simple logistic regression, we repeated our multivariable analysis and entered only those predictors that were statistically significant in univariate analyses. This procedure also enabled us to be parsimonious with the predictors entered and eventually prevented from overfitting of the model. 

The concerning statistics and results sections have been revised accordingly:

Page 7, lines 138-143: We used binary logistic regression to model the crude prediction of RSV and influenza virus infection for the probability of ICU admission or death within the first 7 days after hospital admission. We then applied multivariable models to adjust for additional predictors of our primary outcome in patients with RSV compared to influenza virus infection. Only predictors that were statistically significant in univariate analyses were considered in the multivariable model.

Page 10, lines 192-199: In the crude analysis, RSV patients were twice as likely to be admitted to ICU or die within 7 days after admission compared to patients with influenza (OR 2.06, 95% CI 1.09-3.90, p=.027) (Table 3). However, the association was no longer significant after controlling for temperature, leukocyte count, and presence of a pulmonary infiltrate on admission. While neither fever nor the underlying respiratory virus appeared to have a significant impact on the primary outcome in the adjusted analysis, a higher leukocyte count (adjusted OR 1.07, 95% CI 1.02-1.13) and the presence of a pulmonary infiltrate on chest X-ray (aOR 3.41, 95% CI 1.93-6.02, p<.001) appeared to significantly increase the risk of an unfavourable outcome.

6. Presence of pulmonary infiltrate has been found to increase the risk of ICU admission or mortality. This was presented as a significant finding and discussed briefly. However, this finding does not seem to align with the objectives of the study of comparing disease severity of RSV vs influenza.

In the discussion, the authors “suspect the influence of pneumonic infiltrate was so strong that age and comorbidities were no longer relevant..”. From Table 2, it looks like the effects of pneumonic infiltrate attenuating the effect of the RSV instead of these potential confounders. The authors could relate these findings better by further investigating the proportion of patients who experienced the event and had pulmonary infiltrates, stratified by RSV/influenza.

Response: We agree with the comment that the finding does not align with the study objectives and thank the reviewer for suggesting an additional analysis. We now performed a subgroup analysis, evaluating the impact of age on ICU admission or death within 7 days after admission in patients presenting with a pulmonary infiltrate stratified by influenza and RSV. A total of 29 (25.2%) influenza cases and 7 (21.9%) RSV cases in this subgroup experienced the primary outcome. The difference was not statistically significant. However, the proportion of cases reaching the outcome between the influenza and RSV group clearly differed by age category with a higher proportion of elderly (over 80 years of age) patients in the RSV group. The numbers were too small to perform any meaningful statistics. We suggest adding this figure to the results and remove the original figure 2 as suggested by reviewer #2.

The revised section reads accordingly:

Page 10, lines 199-202: Patients with pneumonia on admission were 3.41 times more likely to be transferred to ICU or to die within the first 7 days than patients without pneumonia (aOR 3.41, 95% CI 1.93-6.02, p<.001). The addition of this predictor to the model, however, seemed to attenuate the effect of RSV as indicated by a decrease in the adjusted OR. 

Page 11, lines 208-216: To better understand the association of a pulmonary infiltrate with clinical outcome, we performed a subgroup analysis, examining the impact of age on ICU admission or death within 7 days after admission in patients presenting with a pulmonary infiltrate stratified by influenza and RSV. First, we found that 29 of 115 (25.2%) influenza cases and 7 of 32 (21.9%) RSV cases in this subgroup experienced the primary outcome. The difference was not statistically significant. However, we note that the proportion of cases reaching the outcome between the influenza and RSV group clearly differed by age category with a higher proportion of elderly (over 80 years of age) patients in the RSV group (Fig 2). The identified virus modified the effect of age on the development of the outcome of ICU admission or death among those with a pulmonary infiltrate.

Reviewer #2: Minor Comments:

Methods:

• The influence of seasonality (season I vs. season II) on the outcome of the patients is not mentioned. It would be interesting to explore if the RSV/Influenza-associated hospitalizations varied substantially during the study period, related to the particular virus types and subtypes in circulation

Response: According to your suggestion we evaluated the influence of seasonality. There were significant differences in the proportion of admissions comparing the two respiratory seasons. We have added the numbers of admission for RSV and influenza separated for the 2 respiratory seasons in Table 1. The percentages of patients in the RSV- and influenza-group experiencing the outcome were surprisingly constant over the two seasons. We suggest adding these results to table 2. 

ICU or death within 7 days according to season RSV Influenza p-value a

Admitted in season I, n of total (%) 5/27 (18.5) 25/270 (9.3) 0.128

Admitted in season II, n of total (%) 10/52 (19.2) 23/199 (11.6) 0.145

a p-values were calculated using Chi-square for comparison of proportions.

Unfortunately, we did not collect the data on influenza subtypes. We therefore could not perform such additional analysis which would have been beyond the scope of this manuscript.

Results:

• The authors stated that the proportion of patients with infiltrate on chest X-ray was significantly higher in the RSV group (46.4% vs. 29.9%). However, figure 1 shows 40.5% vs.24.5%. Please revise

Response: We thank Reviewer #2 for detecting this oversight. The percentages have been corrected in the abstract and in Table 1. 

• Primary endpoint is part of the main objective of this paper but the results are not shown in main body. Please move Suppl. Table 1. to main body

Response: This is an excellent suggestion, and we would like to thank you and Reviewer #1 for pointing this out. We moved the outcomes table in the main manuscript as Table 2.

• Justify why the covariates tested in the multivariate model were selected (e.g. Significance criteria/supported by previous literature)

Response: we would like to thank you and Reviewer #1 for highlighting this issue. We repeated the complete analysis. We kindly ask you to refer to our response above provided to Reviewer #1 under point 5.

Reviewer #3:

1. As the inpatient follow up was limited to 7 days, would the authors be expected to fully describe the clinical outcome of adults patients hospitalised with RSV or influenza infection? This is important as "outcomes" is present in both the full and short title of the manuscript.

Response: This is an excellent suggestion, and we would like to thank you and the other reviewers for pointing this out. We moved the outcomes table in the main manuscript as Table 2.

2. Is Figure 2 necessary? In its current form, it appears to add little to the manuscript as the cutoff is at 7 days

Response: Again, this is an excellent remark. We removed this figure. However, since Reviewer #1 recommended performing a subgroup analysis, we propose to add this figure from the subgroup analysis evaluating age as a predictor of the primary outcome in the subgroup of those with a pulmonary infiltrate stratified by the underlying viral disease. 

3. Please review reference 16 - this research article had described workflow parameters and in its results, it did not describe sensitivity or specificity of the test referred to.

Response: We thank the reviewer for pointing out this oversight. The correct reference (Zou Xiaohui et al. Int J Infect Dis. 2019;80:92-7.) which is now reference 18. and two additional references (16. Cohen et al. J Clin Microbiol. 2018;56(2) and 17. Ling et al. J Clin Microbiol. 2018;56(3)) which are related to the validation of this test have been added to the manuscript. 

4. Would the authors consider expanding their discussion to other points including clinical prediction models for RSV, possible preventive measures, nosocomial transmission, possible planned clinical interventions for RSV.

Response: We agree with reviewer #3 that the importance of preventive measures and clinical interventions to prevent nosocomial transmission of RSV cannot be emphasized enough. We have added a statement on the importance of virological diagnosis and prevention.

---

## [Decision Letter · Decision Letter 1]

10 May 2021

PONE-D-21-03134R1

Clinical outcomes of adults hospitalized for laboratory confirmed respiratory syncytial virus or influenza virus infection .

PLOS ONE

Dear Dr. Vuichard-Gysin,

Thank you for submitting your manuscript to PLOS ONE. After careful consideration, we feel that it has merit but does not fully meet PLOS ONE’s publication criteria as it currently stands. Therefore, we invite you to submit a revised version of the manuscript that addresses the points raised during the review process.

We look forward to receiving your revised manuscript.

Kind regards,

Tai-Heng Chen, M.D.

Academic Editor

PLOS ONE

Journal Requirements:

Reviewers' comments:

Reviewer's Responses to Questions

**Comments to the Author**

1. If the authors have adequately addressed your comments raised in a previous round of review and you feel that this manuscript is now acceptable for publication, you may indicate that here to bypass the “Comments to the Author” section, enter your conflict of interest statement in the “Confidential to Editor” section, and submit your "Accept" recommendation.

Reviewer #1: All comments have been addressed

2. Is the manuscript technically sound, and do the data support the conclusions?

Reviewer #1: Yes

3. Has the statistical analysis been performed appropriately and rigorously? 

Reviewer #1: Yes

4. Have the authors made all data underlying the findings in their manuscript fully available?

Reviewer #1: Yes

5. Is the manuscript presented in an intelligible fashion and written in standard English?

Reviewer #1: Yes

6. Review Comments to the Author

Reviewer #1: I thank the authors for revising their manuscript to make it clearer and transparent. Responses are also well organized inside the Response to Reviewers file. This research shows that it is not easy differentiating adverse clinical outcomes between viral pathogens, especially on viruses that are detected less frequently.

Major comment:

With this revised analysis, the authors could not find any difference in severity between RSV and influenza. However, higher leucocyte counts, and presence of chest infiltrates increased the risk of ICU admission/death. Importantly, RSV patients also had higher median leucocyte counts and number of chest infiltrates. These baseline differences make any inference attributable to the pathogen problematic. More needs to be done in order to understand the relationship between the 2 predictors of interest and RSV/influenza.

It will be useful for the authors to give a plausible explanation for their findings and discuss the role of WBC and chest infiltrates in viral etiology, especially if others have found similar clinical characteristics between RSV and influenza. What is the possible pathway here? Eg: (RSV -> higher leucocyte -> ICU/death or RSV -> higher leucocyte -> ICU/death). Discussing this allow the readers to understand if these findings are biologically plausible or due to self-selection factors (ie: RSV patients coming into the ED at a sicker phase of illness).

Minor comments:

1. In Study Setting and Population, “who’s condition required hospital admission.” Should be whose.

2. In Data Collection: week 44 to 16, did you mean epi weeks or calendar week? Could just specify.

7. PLOS authors have the option to publish the peer review history of their article (what does this mean?). If published, this will include your full peer review and any attached files.

Reviewer #1: No

---

## [Author Response · Author response to Decision Letter 1]

25 May 2021

We would like to thank all reviewers again for their efforts and are pleased that our manuscript comes close to meeting the PLOS ONE publication criteria. Thank you for inviting us to submit a revised manuscript. We hope that we were able to adequately address the points raised.

Please find below our response to reviewer #1.

Reviewer #1: Major comment:

With this revised analysis, the authors could not find any difference in severity between RSV and influenza. However, higher leucocyte counts, and presence of chest infiltrates increased the risk of ICU admission/death. Importantly, RSV patients also had higher median leucocyte counts and number of chest infiltrates. These baseline differences make any inference attributable to the pathogen problematic. More needs to be done in order to understand the relationship between the 2 predictors of interest and RSV/influenza.

It will be useful for the authors to give a plausible explanation for their findings and discuss the role of WBC and chest infiltrates in viral etiology, especially if others have found similar clinical characteristics between RSV and influenza. What is the possible pathway here? Eg: (RSV -> higher leucocyte -> ICU/death or RSV -> higher leucocyte -> ICU/death). Discussing this allow the readers to understand if these findings are biologically plausible or due to self-selection factors (ie: RSV patients coming into the ED at a sicker phase of illness).

Authors’ response: 

We agree that the association between higher leukocyte count/presence of chest infiltrate and ICU admission or death disserves an in-depth discussion. After reconsidering our findings and looking again into the literature, we feel that both considerations remain speculative.

We suggest the following explanation to be added to the discussion section (pages 11-12, line 228-247):

“There was a statistically significantly higher proportion of leukocytosis among patients with RSV compared to those with influenza, albeit with most patients having only mild leukocytosis at most. In multivariable analysis, leukocytosis but not RSV was associated with more severe clinical course. The association between RSV and leukoyctosis has received some attention recently. A study of 243 US adults with RSV showed that 11% had leukocytosis with no difference between those with moderate to severe disease (10%) and those with milder illness (12%, p=0.8). In this study, patients with more severe disease had more frequently pneumonia (Belongia et al. OFID 2018). In a large Chinese study of 1046 adults with RSV, 38% had leukocytosis overall including 44% of those with influenza coinfection while even with these large case numbers the authors were unable to find an association between leukocytosis and severity or radiologic infiltrates (Cui et al. Plos One 2016). In children with RSV, one study found leukocytosis in 24% of those with fever and 18% of those without fever. In those with fever, there was also a higher likelihood of bacterial infection, but among febrile children with a white blood count (WBC) count below 30,000 per milliliter, there was no difference in the WBC between those with and those without bacterial superinfection (Purcell et al. PIDJ 2007). Therefore, it remains speculative, whether the observed higher WBC count in patients with RSV compared to influenza in our study is related to more frequent bacterial superinfections (antibiotic therapy had a non-significantly higher point estimate in RSV than influenza) or to is a marker of more severe clinical presentation of a viral infection, regardless of whether RSV or influenza. WBC was previously shown to have similar prognostic accuracy as CRP for severe outcome in community acquired pneumonia but without being sensitive or specific enough to work as a predictor on its own (Christ-Crain et al. Crit. Care 10, R96 2006).”

For better transition, we also slightly adjusted the first sentence in the subsequent paragraph (page 12, lines 248-250):

“In contrast to previous results that pointed towards older patients and patients with comorbidities (4, 10), our primary findings of serious outcomes in patients with RSV were independent of age and comorbidities.” 

Reviewer #1: Minor comments:

1. In Study Setting and Population, “who’s condition required hospital admission.” Should be whose.

Reply: Thank you very much. We have corrected this.

2. In Data Collection: week 44 to 16, did you mean epi weeks or calendar week? Could just specify.

Reply: Thank you very much. We considered calendar weeks. We clarified this within the text (page 6, line 104-105): 

“This prospective surveillance starts in calendar week 44 in the previous year and extends to calendar week 16 in the following year.”

---

## [Decision Letter · Decision Letter 2]

31 May 2021

Clinical outcomes of adults hospitalized for laboratory confirmed respiratory syncytial virus or influenza virus infection .

PONE-D-21-03134R2

Dear Dr. Vuichard-Gysin,

We’re pleased to inform you that your manuscript has been judged scientifically suitable for publication and will be formally accepted for publication once it meets all outstanding technical requirements.

Kind regards,

Tai-Heng Chen, M.D.

Academic Editor

PLOS ONE

Reviewers' comments:

Reviewer's Responses to Questions

**Comments to the Author**

1. If the authors have adequately addressed your comments raised in a previous round of review and you feel that this manuscript is now acceptable for publication, you may indicate that here to bypass the “Comments to the Author” section, enter your conflict of interest statement in the “Confidential to Editor” section, and submit your "Accept" recommendation.

Reviewer #1: All comments have been addressed

2. Is the manuscript technically sound, and do the data support the conclusions?

Reviewer #1: Yes

3. Has the statistical analysis been performed appropriately and rigorously? 

Reviewer #1: Yes

4. Have the authors made all data underlying the findings in their manuscript fully available?

Reviewer #1: Yes

5. Is the manuscript presented in an intelligible fashion and written in standard English?

Reviewer #1: Yes

6. Review Comments to the Author

Reviewer #1: (No Response)

7. PLOS authors have the option to publish the peer review history of their article (what does this mean?). If published, this will include your full peer review and any attached files.

Reviewer #1: No

---

## [Editor Report · Acceptance letter]

14 Jul 2021

PONE-D-21-03134R2 

Clinical outcomes of adults hospitalized for laboratory confirmed respiratory syncytial virus or influenza virus infection. 

Dear Dr. Vuichard-Gysin:

I'm pleased to inform you that your manuscript has been deemed suitable for publication in PLOS ONE. Congratulations! Your manuscript is now with our production department. 

Kind regards, 

on behalf of

Dr. Tai-Heng Chen 

Academic Editor

PLOS ONE